# Near-infrared uncaging or photosensitizing dictated by oxygen tension

Erin D. Anderson[1], Alexander P. Gorka[1] & Martin J. Schnermann[1]

Existing strategies that use tissue-penetrant near-infrared light for the targeted treatment of cancer typically rely on the local generation of reactive oxygen species. This approach can be impeded by hypoxia, which frequently occurs in tumour microenvironments. Here we demonstrate that axially unsymmetrical silicon phthalocyanines uncage small molecules preferentially in a low-oxygen environment, while efficiently generating reactive oxygen species in normoxic conditions. Mechanistic studies of the uncaging reaction implicate a photoredox pathway involving photoinduced electron transfer to generate a key radical anion intermediate. Cellular studies demonstrate that the biological mechanism of action is $O_2$-dependent, with reactive oxygen species-mediated phototoxicity in normoxic conditions and small molecule uncaging in hypoxia. These studies provide a near-infrared light-targeted treatment strategy with the potential to address the complex tumour landscape through two distinct mechanisms that vary in response to the local $O_2$ environment.

[1] Chemical Biology Laboratory, Center for Cancer Research, National Cancer Institute, Frederick, Maryland 21702, USA. Correspondence and requests for materials should be addressed to M.J.S. (email: martin.schnermann@nih.gov).

The development of methods that use tissue penetrant near-infrared light to treat diseases such as cancer is a long-standing goal. Existing clinical modalities rely on photodynamic therapy (PDT) approaches, which use light to generate toxic levels of reactive oxygen species (ROS)[1]. An inherent consequence is a reliance on substantial $O_2$ tension. However, tumour microenvironments are frequently hypoxic, which can be exacerbated by $O_2$-consuming PDT treatment[2]. Consequently, the requirement of $O_2$ imposes a significant limitation in many cases[3]. One appealing alternative is to use near-infrared light to deliver bioactive molecules, thereby generating a spatially targeted biological effect while avoiding the undesired side effects associated with systemic delivery. Progress using metallic nanoparticles, liposomal formulations, and, recently, small molecule uncaging has been achieved[4–14]. However, the site-specific release of bioactive molecules alone sacrifices the benefits of conventional PDT treatments (that is, catalytic light-driven generation of toxic ROS). Here we describe a strategy where the effect of near-infrared light switches between small molecule uncaging and ROS-generation in an $O_2$ tension-dependent manner.

This approach is based on the reactivity of silicon phthalocyanines (SiPcs). Specifically, we predicted a suitably substituted SiPc would undergo axial ligand exchange in low-$O_2$ conditions, to be used for hypoxia-selective drug delivery, while generating toxic ROS under aerobic conditions (Fig. 1a). The subject of extensive photophysical characterization, various SiPcs have been shown to undergo efficient intersystem crossing and exhibit long-lived triplet states[15]. Consequently, singlet oxygen ($^1O_2$) is generated in useful quantum yields, which has led to extensive exploration of SiPcs for PDT applications. In fact, several examples have progressed in preclinical and clinical contexts[15–17]. In contrast to the detailed photophysical characterization, the photochemical reactivity of SiPcs has received less attention. The cleavage of SiPc–carbon bonds (bond dissociation energy $\approx 40\,\text{kcal mol}^{-1}$) (ref. 17), first observed in synthetic studies,

was recently used to cleave the SiPc chromophore from a gold nanoparticle[18–21]. The focus of the present study is the cleavage of more stable SiPc-oxygen bonds (bond dissociation energy $\approx 90\,\text{kcal mol}^{-1}$) (ref. 18), which was only investigated in a single prior report[22]. In these studies, it was demonstrated that oxygen (phenol, carboxylate, sulfonate and silanol) ligands of symmetrical octaphenoxy SiPcs can be cleaved with 690 nm light in useful quantum yields. However, the photochemical reaction was carried out only in dimethylsulphoxide (DMSO), presumably due to the poor aqueous solubility of the SiPcs examined, and it was unknown if this unique reactivity could be translated to a biological setting. Moreover, the role of the unusual octaphenoxy substituents, and the mechanism of this cleavage process, remained to be defined.

The model that guided the development of this approach is shown in Fig. 1b. The basis for the conditional role of $O_2$ on reaction course is the ability of a long-lived triplet state to undergo either energy transfer or photoinduced electron transfer (PET) processes[23,24]. The former proceeds readily in oxygenated environments to generate toxic $^1O_2$ (ref. 25). With regard to the latter, photochemical reactions involving PET have been explored for small molecule uncaging using visible light[10,26–29]. Strategies that achieve both ligand cleavage and $^1O_2$ generation have been illustrated with ruthenium-based complexes[30,31]. Prior work in the area of photochemical energy applications has characterized a variety of single-electron photochemical processes involving phthalocyanine macrocycles[23,32]. As a consequence, it appeared reasonable that SiPc oxygen ligand exchange would entail radical anion formation followed by ejection of the anionic oxygen ligand. As many triplet state-mediated events are sensitive to $O_2$, including mechanistically distinct PET processes involving phthalocyanines, ligand exchange was predicted to occur preferentially in low-$O_2$ conditions[25,33,34]. A final aspect of this mechanism is the requirement of an electron donor. Biological thiols such as glutathione (GSH) were predicted to serve in this role because they are present in the intracellular milieu in high concentration (2–10 mM) and exhibit suitable redox potentials[35]. These studies describe mechanistic analyses of the SiPc photoredox reaction under biologically relevant aqueous conditions. Applying this approach to a cellular context, we demonstrate that the biological mechanism of action is $O_2$-tension dependent, with small molecule uncaging effects in hypoxia and ROS-dependent toxicity in normoxic conditions. These studies extend PET uncaging into the near-infrared range and illustrate that a PET-based reaction can enable hypoxia-selective drug delivery.

## Results

**Design and synthesis.** The use of axially unsymmetrical SiPcs allows for the appendage of both a payload and a solubilizing ligand. To carry out mechanistic experiments, we prepared compound **2** (Table 1), which contains the fluorescence reporter 4-methylumbelliferone as an axial ligand. To investigate the application of these molecules for small molecule drug delivery, we have prepared **3** and **4**. These compounds are substituted with the potent tubulin polymerization inhibitor, combretastatin-A4 (Z-CA4), or its much less biologically active E-isomer (E-CA4), respectively[36,37]. As the solubilizing ligand, we have used 1,3-bis(trimethylaminium)-2-propanol, which had been shown to enhance the aqueous solubility of other SiPcs[38].

The synthesis of these molecules entails a three-step sequence from commercial SiPcCl₂, and two steps from previously known **1** (Table 1)[38]. After optimization, mono-ligand exchange from **1** proceeded in excellent conversion with phenol nucleophiles. Thus, compound **1** was exposed to 2–4 equivalents of the phenol in CHCl₃ to provide the intermediate tertiary diamines in

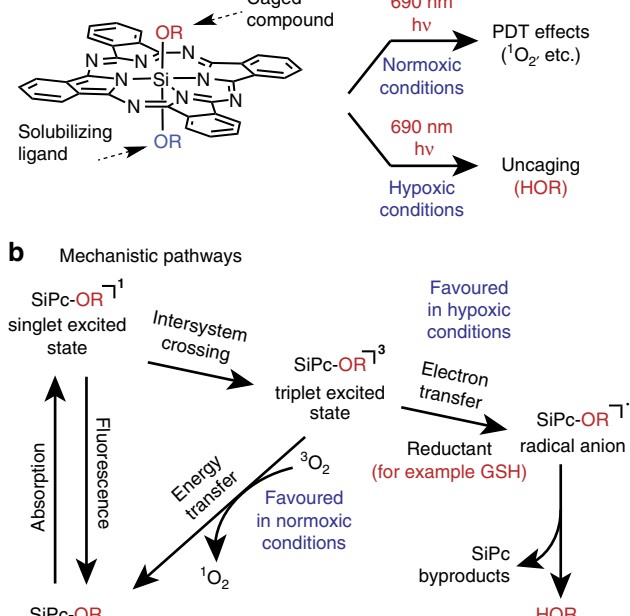

**Figure 1 | Design of approach. (a)** $O_2$ tension-dependent mechanism of action. **(b)** Mechanistic pathways leading to ROS generation or small molecule uncaging.

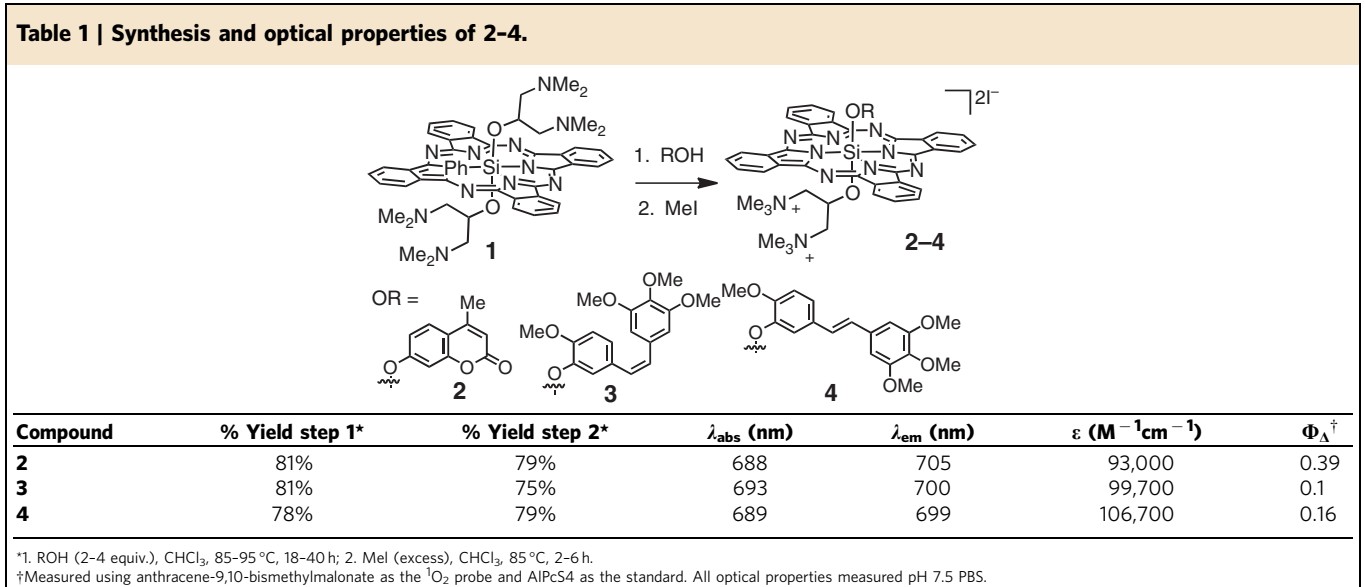

**Table 1 | Synthesis and optical properties of 2–4.**

| Compound | % Yield step 1* | % Yield step 2* | $\lambda_{abs}$ (nm) | $\lambda_{em}$ (nm) | $\varepsilon$ (M$^{-1}$cm$^{-1}$) | $\Phi_\Delta$† |
|---|---|---|---|---|---|---|
| 2 | 81% | 79% | 688 | 705 | 93,000 | 0.39 |
| 3 | 81% | 75% | 693 | 700 | 99,700 | 0.1 |
| 4 | 78% | 79% | 689 | 699 | 106,700 | 0.16 |

*1. ROH (2–4 equiv.), CHCl$_3$, 85–95 °C, 18–40 h; 2. MeI (excess), CHCl$_3$, 85 °C, 2–6 h.
†Measured using anthracene-9,10-bismethylmalonate as the $^1O_2$ probe and AlPcS4 as the standard. All optical properties measured pH 7.5 PBS.

78–81% yield after purification on alumina. Methylation (MeI, 85 °C, CHCl$_3$) followed by recrystallization afforded the final products in 75–79% yield. Compounds **2–4** are monomeric in aqueous solution (Supplementary Fig. 1) and exhibit absorption and emission maxima similar to other SiPcs (Table 1, for full curves see Supplementary Fig. 2). We confirmed that these compounds were effective photosensitizers using a relative method with anthracene-9,10-bismethylmalonate as the $^1O_2$-specific probe and the tetrasulfophthalocyanine complexes of aluminum (AlPcS4) as the standard. The quantum yields of $^1O_2$ generation ($\Phi_\Delta$) of **2**, **3**, and **4** are similar to those of other aqueous-soluble SiPcs, although the values for **3** and **4** are lower than for **2** (refs 33,39,40).

**Uncaging analysis.** SiPc **2**, which represents a caged form of the fluorescence reporter 4-methylumbelliferone, was used to investigate the uncaging reaction (Fig. 2a). Irradiation was carried out with a convenient 690 nm (± 20 nm) light-emitting diode (LED) source at 20 mW cm$^{-2}$, which was used throughout the studies described below. We found that 690 nm irradiation of a deoxygenated 25 μM solution of **2** in pH 7.5 PBS buffer with 5 mM GSH for 10 min led to a dramatic increase (~100-fold) in the characteristic 4-methylumbelliferone fluorescent signal ($\lambda_{ex} = 360$ nm, $\lambda_{em} = 445$ nm). This process was contingent upon the presence of light, GSH and deoxygenation with bubbled Ar (Fig. 2b). We have verified that the increase in fluorescence derives from free umbelliferone by evaluating its yield using high-performance liquid chromatography (HPLC) (71 ± 4%). Other electron donors, such as ascorbic acid, cysteine, and even HEPES buffer, also induced this reaction to varying degrees, with the extent of axial ligand release being roughly dependent on the reduction potential of the electron donor (Supplementary Fig. 3). We have also investigated the effect of initial O$_2$ concentration on uncaging. A range of physiologically relevant initial O$_2$ levels (5, 10, 20 and 40 mmHg)[41] were established using Ar deoxygenation. The level of 4-methylumbelliferone release depends on both initial O$_2$ concentration and irradiation time. Efficient release was observed from the samples with an initial O$_2$ level of 20 mmHg or less after a 12 min irradiation (Supplementary Fig. 4). Of note, unlike in the prior studies using octaphenoxy-substituted SiPcs[22], uncaging of 4-methylumbelliferone from **2** did not proceed efficiently in DMSO with 690 nm light alone. Thus, it appears appending octaphenoxy

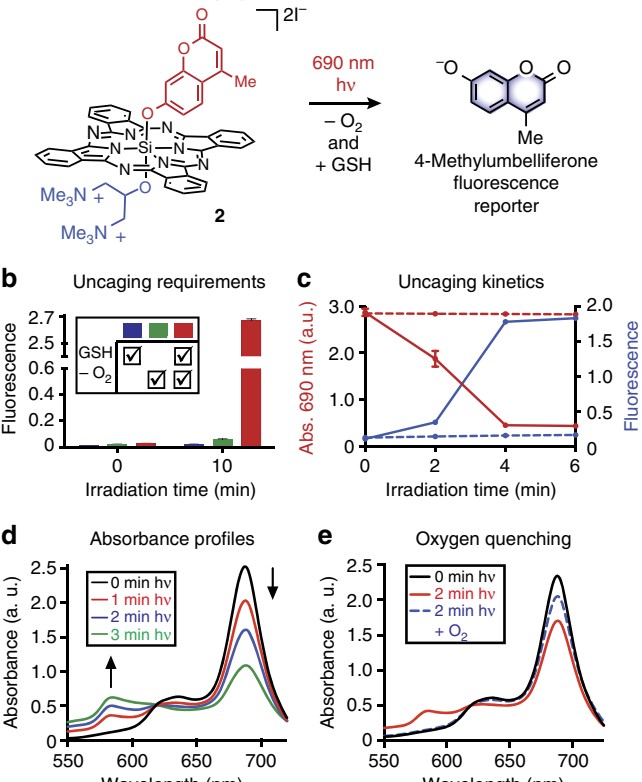

**Figure 2 | Uncaging Analysis.** (**a**) General scheme. (**b**) Fluorescence intensity at 445 nm of a 25 μM solution of **2** (PBS, pH 7.5) with 5 mM GSH, no deoxygenation (blue); no GSH, deoxygenation (green); or 5 mM GSH, deoxygenation (red) at 0 and 10 min exposure to 20 mW cm$^{-2}$ 690 nm light. (**c**) Absorbance at 690 nm (red) and fluorescence at 445 nm (blue) of a deoxygenated 25 μM solution of **2** (PBS with 5 mM GSH, pH 7.5) with (solid line) or without (dashed line) 20 mW cm$^{-2}$ 690 nm irradiation. (**d**) Absorbance profiles resulting from 0, 1, 2 or 3 min irradiation of the solution used in **c**. (**e**) Absorbance profiles resulting from 0 or 2 min irradiation or 2 min irradiation followed by introduction of O$_2$ to a solution identical to that used in **c** and **d**. Error bars represent the standard deviation resulting from triplicate measurement.

ligands to the phthalocyanine macrocycle impacts the axial ligand cleavage process.

The impact of the uncaging reaction on the SiPc macrocycle was examined. Irradiation (690 nm, 20 mW cm$^{-2}$) of a deoxygenated solution of **2** containing GSH (5 mM) induces the loss of the SiPc Q-band at 690 nm with similar kinetics to the appearance of the reporter signal (Fig. 2c). Reductive, anaerobic conditions had a significant effect on the quantum yield of SiPc photodegradation ($\Phi_{PD}$), which is defined as (molecules photobleached)/(photons absorbed during this time interval). A solution of **2** in deoxygenated PBS containing 5 mM GSH was estimated to have a $\Phi_{PD}$ of $2.7 \times 10^{-3}$, roughly 900-fold greater than in PBS alone without deoxygenation (Supplementary Table 1). With regard to the final identity of the SiPc macrocycle, the irradiation process leads to the formation of an insoluble blue precipitate (Supplementary Fig. 5). This material may derive from monoligated or aggregated phthalocyanine products, both of which are known to exhibit limited aqueous solubility[22,42].

Analysis of the absorbance spectra provides strong evidence for the role of radical anion intermediates in the uncaging pathway. Prior studies characterizing PET reactions of SiPcs and other phthalocyanines have established that the appearance of a sharp peak at ~585 nm is diagnostic of a phthalocyanine macrocycle radical anion. These radical anion species have been generated through either electrochemical reduction[43,44] or photoredox[33,45] methods, and, critically, this peak is distinct from that of the phthalocyanine radical cation[43]. In our studies, we observe a sharp increase in absorbance at 585 nm concomitant with Q-band decrease (Fig. 2d). The appearance of this peak is dependent on light, GSH and O$_2$ exclusion (Supplementary Fig. 6). The peak is relatively long-lived in the dark, with a $t_{1/2}$ of 45 s (Supplementary Fig. 7). As O$_2$ can often efficiently quench radical anions, we investigated if that was the case here[32,33]. We found that the discrete peak at 585 nm is lost immediately upon introduction of O$_2$ to a deoxygenated solution, with partial recovery of the 690 nm Q-band (Fig. 2e). The extent of this O$_2$-induced Q-band recovery is a function of irradiation time (Supplementary Table 2).

Several additional observations support the general reaction model. With regard to the effect of O$_2$, most examples of well-characterized SiPc PET processes exclude O$_2$ in the experimental protocol. The exclusion of O$_2$ has been shown to increase the lifetime of both triplet state and radical anion intermediates to favour such pathways[32,45]. While most cases of PET reactions involve the triplet state[23], there are cases where amines have been shown to reduce the singlet excited state of SiPcs[33,44]. We investigated the ability of GSH to interact with the singlet state of SiPc **2** through a Stern-Volmer analysis of SiPc fluorescence emission with varying concentrations of GSH. Increased GSH concentration had no effect on fluorescence emission, suggesting that PET does not occur with the singlet state (Supplementary Fig. 8). This data provides further support for the notion that the triplet state is a key intermediate en route to ligand exchange. Lending additional support to the reaction model, the Rehm-Weller estimation has been used to predict that the reduction potential of disubstituted SiPcs increases from a ground state range of $E_{red} = -0.9$ to $-0.7$ V versus SCE to a range of $E_{red}^* = +0.35$ to $+0.56$ V versus SCE in the triplet excited state[45]. With these values, reduction with thiol electron donors such as GSH ($E_{red} = -0.50$ versus SCE) becomes favourable[46]. While further analysis will likely provide additional insight, the results above support the model shown in Fig. 1 and convinced us to pursue cellular studies.

**Cellular studies.** We sought to examine if an O$_2$ tension-dependent shift from ROS-mediated toxicity to drug release

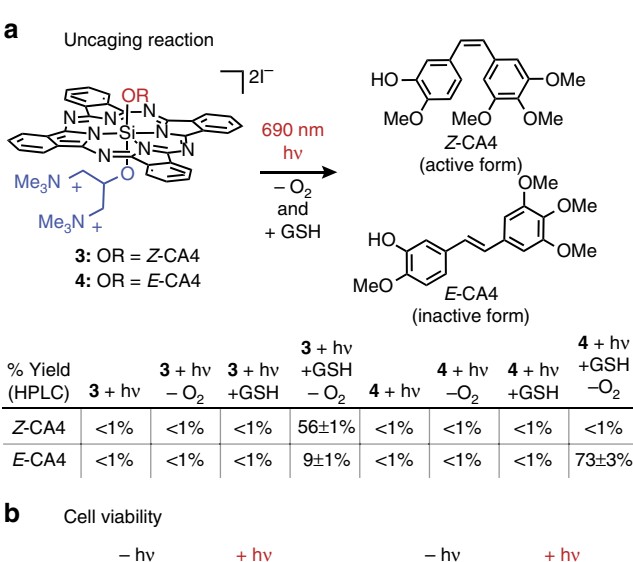

**a** Uncaging reaction

| % Yield (HPLC) | 3 + hv | 3 + hv − O₂ | 3 + hv +GSH | 3 + hv +GSH − O₂ | 4 + hv | 4 + hv −O₂ | 4 + hv +GSH | 4 + hv +GSH −O₂ |
|---|---|---|---|---|---|---|---|---|
| Z-CA4 | <1% | <1% | <1% | 56±1% | <1% | <1% | <1% | <1% |
| E-CA4 | <1% | <1% | <1% | 9±1% | <1% | <1% | <1% | 73±3% |

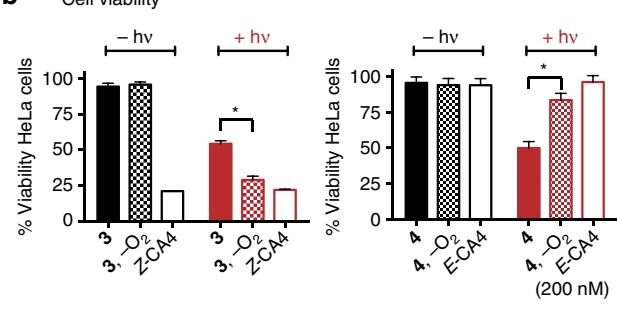

**b** Cell viability

**c** Cell cycle analysis

| | Z-CA4 | E-CA4 | 3 + hv | 3 + hv − O₂ | 4 + hv | 4 + hv − O₂ |
|---|---|---|---|---|---|---|
| % G1 | 1.9% | 67% | 58% | 2.4% | 68% | 50% |
| % G2/M | 87% | 26% | 9.4% | 84% | 9.3% | 27% |
| Ratio $\frac{G2/M}{G1}$ | 46 | 0.4 | 0.2 | 35 | 0.1 | 0.6 |

**Figure 3 | Uncaging of Z-CA4 and E-CA4 from 3 and 4 and cellular effects.** (**a**) General Scheme and yields of Z-CA4 and E-CA4 from **3** and **4**, respectively, in the presence of 690 nm light, hypoxia (bubbled Ar) and GSH (5 mM). A solution of **3** or **4** (25 μM in DMEM with 5 mM GSH) was deoxygenated with Ar, irradiated with 35 J cm$^{-2}$ of 690 nm light, and the yield of ligand released was quantified by HPLC. Experiments were performed in triplicate and reported ± s.e. (**b**) HeLa cell viability in the presence of 200 nM **3**, **4**, Z-CA4 and E-CA4 under normoxic and hypoxic conditions, with and without 690 nm irradiation (50 J cm$^{-2}$). Viability was assessed using a standard MTT assay. Error bars represent the standard deviation resulting from quadruplicate measurements. (*) indicates a statistically significant difference ($P < 0.001$). (**c**) Cell cycle analysis of HeLa cells treated as in **b**. Following treatment (24 h), cells were trypsinized, fixed with 70% ethanol, stained with propidium iodide and analysed for DNA content by flow cytometry.

occurs in a cellular context. To distinguish between ROS-mediated effects and those derived from small molecule release, we compared the biological activity of **3** and **4**. The former is substituted with the potent tubulin polymerization inhibitor Z-CA4, and the latter its much less active E isomer, E-CA4. Thus, under hypoxic conditions, and in the presence of near-infrared light and cellular thiols, we anticipated that both **3** and **4** should similarly release Z- and E-CA4, respectively. However, only in the case of **3** should the effects of tubulin polymerization inhibition be observed. By contrast, drug release should not occur to a significant extent in normoxic conditions with similar ROS-dependent effects observed for both **3** and **4**.

Initial characterization investigated the ability of **3** and **4** to release Z-CA4 and E-CA4, respectively. We found that with deoxygenation, GSH and 690 nm irradiation of **3** and **4** provided Z-CA4 and E-CA4 in 56 and 73% yields, respectively, albeit requiring somewhat higher light doses than needed with **2** ($\sim 35\,\mathrm{J\,cm^{-2}}$ for **3** and **4** versus $\sim 15\,\mathrm{J\,cm^{-2}}$ for **2**) to reach full conversion. As above, release was contingent on deoxygenation, the thiol reductant, and light (Fig. 3a). Also of note, a small amount of E-CA4 (9%) is observed from Z-CA4-substituted compound **3** under the uncaging conditions. This result is consistent with the facile and well-documented isomerization of Z-CA4 to E-CA4 (ref. 37). Conversely, only trace Z-CA4 ($<1\%$) is observed from the E-CA4-substituted compound, **4**, under the same conditions. Moreover, these compounds displayed excellent dark stability, as **3** only released trace ($<1\%$) Z-CA4 after 6 h in deoxygenated DMEM supplemented with 5 mM GSH.

Using **3** and **4**, we have carried out a set of cell viability and cell cycle experiments (Fig. 3b,c). Fluorescence microscopy using HeLa cells indicated that these molecules were taken up into cells and appear to localize in lysosomal compartments (Supplementary Fig. 9). Maximal intracellular signal was observed following a 3 h preincubation, therefore, these conditions were used throughout the studies below. Cell viability measurements (MTT assay) demonstrated that **3** and **4** (200 nM) have insignificant effects on cell growth in the absence of light (Fig. 3b), independent of oxygen status. We then evaluated the effects of these compounds upon exposure to $50\,\mathrm{J\,cm^{-2}}$ of 690 nm light under either normoxic or following incubation in hypoxic conditions. To achieve the latter, cells were placed in a chamber under an $N_2$ atmosphere for 1 h before and during photolysis[47]. Compound **3** elicits 80% growth inhibition under hypoxic conditions, which closely matches that of free Z-CA4 (83%). This activity is diminished under normoxic conditions (48%). In contrast, **4** leads to only 13% growth inhibition under hypoxic conditions, while inhibition under normoxic conditions is similar to that observed with **3** (50%). We have investigated the role of light dose on these experiments. Notably, significant cell viability effects are observed upon irradiation of **3** under hypoxic conditions with $10\,\mathrm{J\,cm^{-2}}$ of light (Supplementary Fig. 10). Finally, no effect on cell growth is observed with irradiation alone (Supplementary Fig. 11).

The growth inhibitory activity for **3** under both normoxic and hypoxic conditions, coupled with the minimal effects of **4** under hypoxia, is supportive of a mechanism involving a transition from ROS-dependent toxicity to drug-based effects. Specifically, we hypothesized that the release of active Z-CA4 from **3** is responsible for growth inhibition under hypoxic conditions, while ROS effects predominate under normoxic conditions. We sought to validate this notion by exploiting unique characteristics of the biological effectors. Inhibition of tubulin polymerization by Z-CA4 leads to mitotic arrest, which is characterized by a buildup of tetraploid G2/M DNA[36]. This can be assessed using cell cycle analysis, which employs flow cytometry to sort diploid versus tetraploid cells treated with the DNA-binding dye propidium iodide[48]. Consistent with hypoxia-selective Z-CA4 release, only irradiation of **3** under hypoxic conditions leads to a similar buildup of G2/M DNA (that is, mitotic arrest) as observed with Z-CA4 alone (Fig. 3c and Supplementary Table 3). By contrast, irradiation of **3** under normoxic conditions or **4**, under either condition of $O_2$ tension, did not induce mitotic arrest.

Having illustrated that Z-CA4 can selectively be released in hypoxic environments, we sought to distinguish ROS-dependent effects. To do this, we exploited the limited diffusion distance of the active ROS species, particularly $^1O_2$. Thus, while $^1O_2$ is known to interact only within the cellular microenvironment in which it was produced, we presumed that released Z-CA4, if in

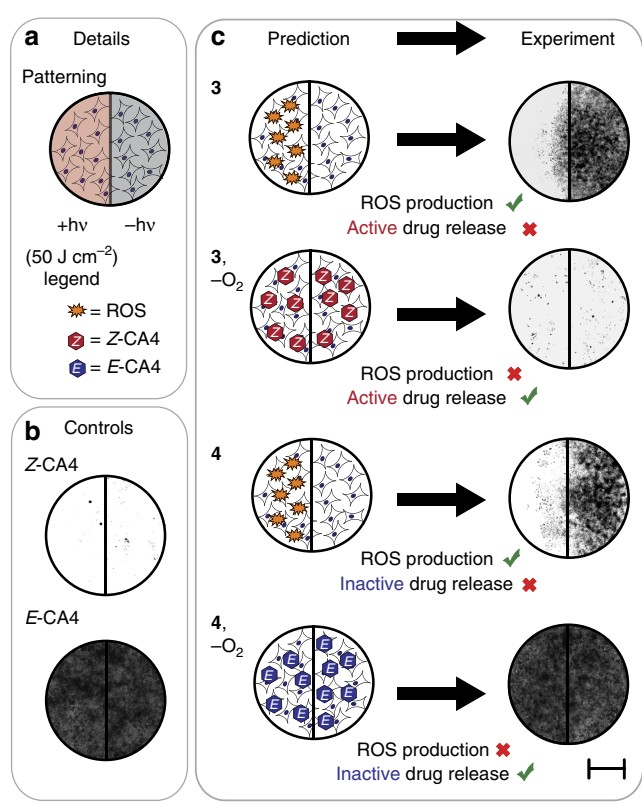

**Figure 4 | Photopatterning studies of 3 and 4 with varied $O_2$ tension.** (**a**) Details—Half of each well was irradiated with 690 nm light ($50\,\mathrm{J\,cm^{-2}}$); the other half was covered with black tape. (**b**) Controls—HeLa cells were treated with 200 nM Z-CA4 and E-CA4 under normoxic and hypoxic conditions and irradiated as indicated above (only the normoxic and irradiated case is shown; the other variants are similar, see Supplementary Fig. 12). Cell viability analysis was performed using MTT after 72 h and wells were imaged under brightfield at $4\times$ magnification. Scale bar, 5 mm. (**c**) Predicted and experimental results of HeLa cells treated with 200 nM **3** and **4** under hypoxic and normoxic conditions and irradiated and analysed as indicated above (for the unirradiated cases, Supplementary Fig. 12).

sufficient concentration, would freely diffuse between the cultured cells before engaging its biological target[49,50]. To assess this, a photopatterning assay was conducted in which only half of each compound well was exposed to 690 nm irradiation (Fig. 4a). In control experiments, Z-CA4 had dramatic effects on cell growth in both sides of the wells, whereas E-CA4 had minimal effects, as visualized by brightfield microscopy of MTT treated cells following a 72 h incubation (Fig. 4b). Under normoxic conditions, **3** and **4** inhibited cell growth in only the irradiated portion of the well (Fig. 4c), consistent with ROS-mediated effects. Under hypoxic conditions, however, **3** led to growth inhibition across the entire well, similar to that of free Z-CA4. By contrast, irradiation of **4** under hypoxic conditions had a minimal effect on cell growth, presumably as a consequence of release of inactive E-CA4. Importantly, in all cases the absence of irradiation had no effect on cell growth in both normoxic and hypoxic conditions (Supplementary Fig. 12). The photopatterning effect was investigated under varied levels of initial $O_2$. With both 1 and 3% $O_2$, cell growth inhibitory effects were observed in the unirradiated area with **3**, but not **4**, albeit somewhat less completely than above (Supplementary Fig. 13). These results suggest that uncaging can occur under the hypoxic conditions found in physiological settings[41,51]. In total, the cell cycle analysis and photopatterning experiments illustrate that growth inhibition from irradiated **3** results from released Z-CA4 under hypoxic

conditions and from photosensitized ROS under normoxic conditions.

## Discussion

These studies have demonstrated that suitably substituted SiPcs effectively undergo 690 nm light-mediated axial ligand exchange to uncage phenols selectively in hypoxic conditions, while generating ROS in normoxic conditions. The photoinduced ligand exchange reaction involves a SiPc radical anion intermediate formed with biologically relevant reductants such as GSH. These studies extend small molecule PET uncaging into the near-infrared range, and illustrate that oxygen tension can dramatically affect uncaging in a cellular context. This approach may serve to address the oxygen requirement of existing PDT treatment methodology. PDT methods involve both energy transfer to generate $^1O_2$ ('type II pathways') and electron transfer to generate superoxide ($^-O_2$) and other radical species ('type I pathways')[1]. The strategy reported here maintains the benefits of type II PDT (that is, the generation of $^1O_2$ in normal $O_2$), but uses electron transfer for small molecule uncaging in a low-$O_2$ environment.

With regard to future applications, the use of a single agent with two distinct mechanisms of action will likely prove valuable for targeting the heterogeneous tumour landscape. This is particularly true because many solid tumours harbour cellular subpopulations characterized by preexisting hypoxia[52]. Consequently, PDT can lead to the delivery of sub-lethal doses of ROS, which have been shown to activate pro-survival mechanisms leading to disease resistant to a variety of treatment approaches[53]. An important objective is to deploy SiPc uncaging with small molecules tailored specifically to these recalcitrant cellular populations. Towards this goal, we are currently pursuing studies that seek to expand the scope of the drug payload and to define, and then harness, this $O_2$ tension-dependent reactivity.

## Methods

**Synthesis and characterization.** The synthesis of **2–4** is described in detail in the Supplementary Methods section. The $^1O_2$ quantum yields were obtained with a relative method using anthracene-9,10-bismethylmalonate as the $^1O_2$-specific probe (data is shown in Supplementary Fig. 14; ref. 54). For HPLC analysis of photolysis reactions, see Supplementary Figs 15–18. For NMR analysis of the compounds in this article, see Supplementary Figs 19–24. For general materials and methods see Supplementary Methods.

**Photolysis experiments.** Samples were irradiated in a quartz cuvette with a septum screw cap using a 690 nm (± 20 nm) LED at 20 mW cm$^{-2}$. Where noted ($-O_2$), samples were deoxygenated by bubbling Ar (balloon) through the septum cap of the sealed cuvette. The internal temperature of the samples did not exceed 25 °C over the irradiation time course. Fluorescence traces were recorded on a fluorimeter. Samples were excited at 360 nm and emission recorded from 400–500 nm to evaluate 4-methylumbelliferone fluorescence. Absorbance spectra were used to evaluate the Q-band ($A_{690}$) and the radical anion ($A_{580}$). The $O_2$ quenching experiments were carried out by introducing bubbled $O_2$ though a balloon. For ligand release experiments, yield was measured by HPLC using an external calibration method.

**Photodegradation quantum yields.** A 3 ml solution of a photosensitizer (20 µM in 50 mM pH 7.5 PBS, ± GSH and ± $O_2$ as noted in Supplementary Table 1) was irradiated at 30 mW cm$^{-2}$ while stirring (stir-bar) in a quartz cuvette. In order to ensure uniform irradiation, a cuvette holder was used to limit the area of light exposure to a $0.8 \times 0.8$ cm$^2$ area. The decrease in 690 nm absorbance was monitored by ultraviolet–visible spectroscopy.

Photodegradation quantum yields were determined by the following equation[55,56]:

$$\phi_{PD} = \frac{VN_A hc\Delta A}{\varepsilon l\lambda P(1 - 10^{-A_0})\Delta t}$$

where $V$ is the volume of the sample (in l), $N_A$ is Avogadro's number, $h$ is Planck's constant, $c$ is the speed of light, $\Delta A$ is the change in absorbance per unit time ($\Delta t$, in s), $\varepsilon$ is the molar absorption coefficient (in M$^{-1}$ cm$^{-1}$), $l$ is the optical pathlength (in cm), $\lambda$ is the excitation wavelength (in m), $P$ is equal to the radiant power (W) as determined by a silicon photodiode photometer, and $(1-10^{-A_0})$ is a correction factor used to account for the amount of transmitted photons.

**Cell photolysis, cytotoxicity and photopatterning.** HeLa (human cervical adenocarcinoma) cells were obtained from American Type Culture Collection (Manassas, VA) and cultured in DMEM supplemented with 4 mM L-glutamine, 25 mM D-glucose, 44 mM sodium bicarbonate, 10% heat-inactivated fetal bovine serum, 100 units ml$^{-1}$ penicillin, 100 µg ml$^{-1}$ streptomycin, and 0.25 µg ml$^{-1}$ amphotericin B. Cells were grown at 37 °C in an atmosphere of 20% $O_2$ and 5% $CO_2$. HeLa cells were seeded into 96-well plates ($5 \times 10^4$ cells per well) and allowed to adhere overnight. Initial seeding densities were such to ensure cells remained in exponential growth for the duration of the assay. Media was replaced with that containing **3**, **4**, Z-CA4, E-CA4 or DMSO at indicated concentrations. Following incubation for 3 h at 37 °C in the dark, media was replaced with inhibitor-free media. For hypoxia experiments, plates were sealed in a modular incubator chamber (Billups-Rothenberg Inc., Del Mar, CA, for images see Supplementary Fig. 25) fitted with a 690 nm LED and the chamber flushed with N$_2$ for 1 h at a flow rate of ~ 15 l min$^{-1}$. For normoxia experiments, plates were exposed to ambient air by removal of the chamber lid. Cells were irradiated with 50 J cm$^{-2}$ of 690 nm light (20 mW cm$^{-2}$ for 42 min) or kept dark. Following photolysis, plates were maintained at 37 °C in the dark for 72 h, after which 20 µl of MTT (3-(4,5-dimethylthiazol-2-yl)-2,5-diphenyltetrazolium bromide) from a 5 mg ml$^{-1}$ stock in PBS was added to each well and incubated for 4 h at 37 °C.

For cytotoxicity measurements, media was removed, 100 µl of DMSO added to each well to solubilize MTT formazan, and absorbance at 550 nm was recorded using a microplate reader. Drug effects were expressed as % cell viability relative to the DMSO (no inhibitor) control. All experiments were conducted in quadruplicate, with error bars representing the standard deviation.

For photopatterning experiments, half of each well was covered with black tape for photolysis. After photolysis and 72 h incubation, media and black tape were removed, cells were treated with MTT as above, and MTT formazan imaged under brightfield at 4 × magnification using an Evos FL Auto microscope (ThermoFisher Scientific). Images in Fig. 4 and Supplementary Figs 12 and 13 result from a composite tile of ten $8,256 \times 6,192$ µm image fields. A circular area of ~ 10 mm circumference about the center of each well is shown (that is, the center 50% of each well). Image processing was conducted with Fiji. All experiments were conducted in quadruplicate, with representative images shown.

**Cell cycle analysis.** HeLa cells were seeded into 96-well plates ($5 \times 10^4$ cells per well) and allowed to adhere overnight. Compound treatments, generation of hypoxia and normoxia, and irradiations were performed as above. Following photolysis, plates were maintained at 37 °C in the dark for 24 h. Cells were harvested by trypsinization and replicate wells pooled to achieve ~ $1 \times 10^6$ cells per condition. Samples were centrifuged for 5 min at 200g, the pellet suspended in $1 \times$ PBS, and centrifugation repeated. The pellet was thoroughly suspended in 0.5 ml of PBS and 4.5 ml of 70% (v/v) ethanol added dropwise with continuous vortexing to fix cells. Fixed samples were stored at 4 °C. Before analysis, samples were warmed to room temperature, centrifuged for 5 min at 200 g, and the ethanol solution thoroughly decanted. The pellet was suspended in 5 ml of PBS, incubated at room temperature for 5 min, and centrifuged as above. The resulting pellet was suspended in 0.5 ml of staining solution (20 µg ml$^{-1}$ propidium iodide, 200 µg ml$^{-1}$ DNase-free RNase A and 0.1% (v/v) Triton X-100 in PBS), incubated for 15 min at 37 °C, 30 min at room temperature, and ~ 15 h at 4 °C. Cells were analysed for DNA content by the Center for Cancer Research (CCR) Flow Cytometry Core (Cancer and Inflammation Program, NCI-Frederick) using a BD Canto II analyser ($\lambda_{ex}$ 488 nm, $\lambda_{em} > 600$ nm). Deconvolution of histograms was performed using ModFit LT (v 4.1.7). A minimum of $1 \times 10^3$ events were modelled per condition. Deconvoluted histograms are shown in Supplementary Figs 26 and 27.

**Data availability.** Data supporting the findings of this study are available within the article and its Supplementary Information files and from the corresponding author upon reasonable request.

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

## Acknowledgements

We thank the Optical Microscopy and Analysis Laboratory (Advanced Technology Program, Frederick National Laboratory for Cancer Research) for assistance in obtaining confocal fluorescence images and the Frederick Center for Cancer Research Flow Cytometry Core (Cancer and Inflammation Program, NCI-Frederick) for help with the cell cycle analysis. We thank Dr Joseph Barchi for NMR assistance and Dr James Kelley for mass spectrometric analysis. Drs Hisataka Kobayashi and Murali Krishna are gratefully acknowledged for assistance with hypoxia experiments and insightful comments. Ms Luxi Qiao is acknowledged for assistance with initial synthetic studies. This work was supported

by the Intramural Research Program of the National Institutes of Health, Center for Cancer Research and the National Cancer Institute, National Institutes of Health.

## Author contributions

E.D.A., A.P.G. and M.J.S. designed the experiments, analysed the data and wrote the manuscript. E.D.A. carried out synthesis and photophysical characterization. A.P.G. conducted the biological experiments.

## Additional information

**Competing financial interests:** The authors declare no competing financial interests.

**Publisher's note**: 

