## [Peer Review File · Nature Communications]

Reviewers' comments:

Reviewer #1 (Remarks to the Author):

The manuscript describes new silicon phthalocyanines (SiPcs) that are able to with their mode of action to induce cell death as a function of oxygen. This is the first type of system of its kind, which is able to induce photo induced cell death in hypoxic tumors via ligand (drug) dissociation as well as via the production of singlet oxygen in normoxic conditions. The authors quantify the two processes, investigate the mechanism, and go on to show the effect on cell cultures. They clearly show the effect of the release of the active tubulin polymerization inhibitor Z-CA4 is associated with cell death upon its release. The manuscript will be of high interest to investigators in the field of cancer therapeutics as well as in the broad area of photochemistry. Publication is recommended following minor revision.

1. References to important, related papers are missing. For example, Kodanko has recently shown the photo induced delivery of cysteine protease inhibitors and this work should be cited as related to "small molecules" (line 32). In addition, reference to the first example of dual ligand dissociation and singlet oxygen production that results in greater cell death should be made (JACS 2014, 136, 17095).

2. Figure 2, Table. The singlet oxygen quantum yields are rather low as compared to typical PDT agents.

3. Lines 129 and 193. These values do not match. The quantum yield for 2 is extremely low. Is there a way to increase it? For the conversion, how long are the compounds irradiated to reach these conversions?

4. Figure 4B. The effect of light alone under similar conditions should be plotted or at least mentioned in the text. Irradiation times should be included.

Reviewer #2 (Remarks to the Author):

The authors describe the photon-dependent release of small molecules from phthalocyanine macrocycles. The manuscript is interesting but lacks some of the central information at this point: The title and the very first paragraph of the manuscript promise "Near-IR Uncaging or Photosensitizing Dictated by Oxygen Tension."

So, the very first data the reader is interested in is the uncaging as a function of oxygen tension and secondly, the photosensitization as a function of oxygen tension. In fact, it would be logical to expect to see the resulting ratio of the yields of uncaging versus photosensitizing as a function of oxygen tension. But this data is never shown. It seems not to be done and really needs to be shown - based on the title and abstract because both, the uncaging and the singlet oxygen formation, have been demonstrated on phthalocyanines individually.

Furthermore, the use of Near-IR light to photocleave a phthalocyanine and produce the PDT drug Pc 4 in water via a radical mechanism has been demonstrated in 2014 by Cheng et al.

Unfortunately, this work is not referenced in this manuscript. Instead a lesser relevant work from the same group from 2015 is referenced as ... "has received much less attention."¹³

That is actually right. It is somewhat of an awkward oversight of this manuscript that this reference is not properly implemented in the introduction of this topic:

"Near infrared light-triggered drug generation and release from gold nanoparticle carriers for

photodynamic therapy" Cheng, Y.; Doane, T. L.; Chuang, C. C.; Ziady, A.; Burda, C. Small (2014), 10(9), 1799-1804.

The paper by Cheng et al is not cleaving Si-O bonds but Si-C bonds but the results and the underlying photophysics are strikingly similar.

Thus, the novelty of this work lies mainly in the switching between uncaging and photosensitizing. As said above to make this then worthy of Nature Comm. The quantification and ratios of the two monitored outcomes needs to be presented.

In addition, the work by Lo et al is cited as example for 1,3-bis(trimethylaminium)-compounds, but in the corresponding reference 30, it turns out that the molecules containing these cationic ligands are less effective compared to other ligands. Is that also true for the presented compounds? What is the singlet oxygen yield of each in normoxic conditions. The values are not presented in this manuscript. From the graph on page S6 they seem to be between 5-10%?

On line 106 of the main text, the authors claim that "The quantum yields of 1O_2 generation (Φ_Δ) of 2, 3, and 4 are similar to those of other SiPcs, when measured in an aqueous environment.²⁵ This is hard to judge in this qualitative comparative statement. Usulan et al. have obtained 11%. The authors should also provide numbers.

One challenge for the reader is that 50J seems to be a lot of light energy for a test tube reaction, and one might ask if this hints to a low photoreaction quantum yield - inefficient release of Z-CA4? From Figure 4 it seems that that Z-CA4 is not photoactive but just a cell toxin. Why then not just deliver Z-CA4, which seems water-soluble, not directly?

There is a typo in "discreet" on line 158.

In summary, there needs to be a more quantitative approach to the description of "Uncaging or Photosensitizing". References and the introduction need to clarify that PDT reagents (such as Pc 4) have already been released in water, as cited above. The work by the Knoer group is a good standard of how much singlet oxygen formation one should expect. It should be highlighted.

The paper needs to be accordingly improved in the presentation of the introduction and the results section and then reconsidered for publication.

Reviewer #3 (Remarks to the Author):

The manuscript presents interesting near IR activated drug (CA4)-phthalocyanine system whose mechanism of action, as either a photosensitizer (singlet oxygen) or a prodrug (releasing active drug), depends on oxygen tension. While the photosensitizing mechanism of silicon phthalocyanine under normoxic condition is well known, the drug release from the conjugate by near IR illumination under oxygen-deprived condition is unique, in particular in cultured cells. Although the Si-O bond cleavage chemistry upon light illumination was reported, its application for NIR-activated drug release moves this chemistry one step forward toward future clinical application. Most of experimental designs are sound and well executed. However, following points need to be addressed before recommended for publication.

1. The authors claim this conjugate system is expected to release CA4 (drug) under the hypoxic condition in tumor. They indeed demonstrated it in test tubes and cultured cells. However, it is not clear whether it really will work as they expect in real tumors because the conditions they used (e.g., oxygen-deprivation condition, (extreme hypoxic condition)) may not close to tumor hypoxia. Quantitative definitions and measures need to be used about "oxygen tension" and "hypoxia" [J Natl Cancer Inst (2001) 93 (4): 266-276]. It will be much clearer if they demonstrate it under oxygen partial pressure of hypoxic tumors. Alternatively, in vivo demonstration of drug release in hypoxic tumor will address this question.

Minor points

2. First page: The term, "targeted", may confuse readers. It seems redundancy because in general focused NIR light itself is used.
3. Line 34: It is not clear what does "these approaches sacrifice the benefits of conventional PDT" mean and how their new approach would solve this problem.
4. Line 205: It is not clear what does "maximal cellular localization" mean. Does it mean maximal uptake to cells or specific subcellular localization to specific organelles in cells?

The reviewer's concerns are addressed with yellow highlighting and changes to the text are indicated with green highlighting.

Reviewers' comments:

Reviewer #1 (Remarks to the Author):

The manuscript describes new silicon phthalocyanines (SiPcs) that are able to with their mode of action to induce cell death as a function of oxygen. This is the first type of system of its kind, which is able to induce photo induced cell death in hypoxic tumors via ligand (drug) dissociation as well as via the production of singlet oxygen in normoxic conditions. The authors quantify the two processes, investigate the mechanism, and go on to show the effect on cell cultures. They clearly show the effect of the release of the active tubulin polymerization inhibitor Z-CA4 is associated with cell death upon its release. The manuscript will be of high interest to investigators in the field of cancer therapeutics as well as in the broad area of photochemistry. Publication is recommended following minor revision.

1. References to important, related papers are missing. For example, Kodanko has recently shown the photo induced delivery of cysteine protease inhibitors and this work should be cited as related to "small molecules" (line 32). In addition, reference to the first example of dual ligand dissociation and singlet oxygen production that results in greater cell death should be made (*JACS* 2014, 136, 17095).

We had initially sought to only include recent references that use light of "near-IR" wavelengths (typically defined as between 650 and 900 nm). However, studies by Kodanko and several others have recently illustrated small molecule advances that use light in the range of 550 nm (green). I have added four additional references to highlight these studies.

The work of Turro is certainly important to this work. We initially chose only to reference a very recent (2015), *Acc Chem Res* review by this author that provides a broader perspective on the same topic, however, given its importance, we have now included the primary reference as well. We also added the following sentence of clarification.

"Strategies that achieve both ligand cleavage and $^1\text{O}_2$ generation have been illustrated with ruthenium based complexes.²³⁻²⁴"

2. Figure 2, Table. The singlet oxygen quantum yields are rather low as compared to typical PDT agents.

This is a good point and caused us to revisit these values carefully through a literature search and by informally consulting with senior colleagues in the field. We have realized using the SOSG method as the $^1\text{O}_2$ -specific probe is not ideal in this case, and rather we should have used the ADMA method. This latter method has been used in three literature examples with a closely related set of SiPc compounds in PBS. Using this method, we obtained a new set of values, albeit with the same general trend seen with SOSG. These values (between 0.38 to 0.10) are very much in line with other compounds in this class. We have added a short sentence describing this changes, as well as a 2 additional references that report similar values.

We confirmed that these compounds were effective photosensitizers using a relative method with anthracene-9,10-bismethylmalonate (ADMA) as the $^1\text{O}_2$ specific probe and the tetrasulfophthalocyanine complexes of aluminum (AlPcS4) as the standard. The quantum yields of $^1\text{O}_2$ generation (Φ_Δ) of 2, 3, and 4 are similar to those of other aqueous-soluble SiPcs, although the values for 3 and 4 are lower than for 2.^{33,39,40}

3. Lines 129 and 193. These values do not match. The quantum yield for 2 is extremely low. Is there a way to increase it? For the conversion, how long are the compounds irradiated to reach these conversions?

It is important to note that the light wavelengths used in our study are far longer (lower energy) than those used for the typical photocages for which quantum yields have been reported. Illustrative are recent reports of BODIPY-based photocages, which use light in the green range (e.g. *J. Am. Chem. Soc.* **2015**, *137*, 3783). In this study, values in the range of 10^{-4} are reported, over an order of magnitude below the values we demonstrate ($\sim 10^{-3}$). With regard to the second question, duration of irradiation is already reported in terms of irradiation times in chemical studies (with reaction completion observed after ~ 5 mins with only LED irradiation, Figure 4C) and as light doses in Joules cm^{-2} for the biological studies. Light dose, which is the total amount of light derived from both flux of light and exposure time, is the standard for biological studies. Our light doses are similar to those used for other small molecule photosensitizers in PDT type applications (when using low nM concentrations of photosensitizers). The supporting information includes actual irradiation times, which, if desired, could be shortened by using stronger laser light sources than the lower energy LEDs we have used.

4. Figure 4B. The effect of light alone under similar conditions should be plotted or at least mentioned in the text. Irradiation times should be included.

We have added a control experiment in which light is applied to cells in isolation and show that no effect on cell viability is observed. This data has been added to the supporting information and a corresponding sentence has been added to the text.

“Finally, no effect on cell growth is observed with irradiation alone (Figure S11).”

Reviewer #2 (Remarks to the Author):

The authors describe the photon-dependent release of small molecules from phthalocyanine macrocycles. The manuscript is interesting but lacks some of the central information at this point:

The title and the very first paragraph of the manuscript promise "Near-IR Uncaging or Photosensitizing Dictated by Oxygen Tension."

So, the very first data the reader is interested in is the uncaging as a function of oxygen tension and secondly, the photosensitization as a function of oxygen tension. In fact, it would be logical to expect to see the resulting ratio of the yields of uncaging versus photosensitizing as a function of oxygen tension. But this data is never shown. It seems not to be done and really needs to be shown - based on the title and abstract because both, the uncaging and the singlet oxygen formation, have been demonstrated on phthalocyanines individually.

The reviewer makes an excellent point. While singlet oxygen generation was illustrated through the reporting of Φ_{Δ} , as described above, the quantification of release vs. O_2 levels is important and useful data. At the time of submission, we did not have the technical capacity to measure O_2 concentration in solution. We since obtained a Piccolo2 Fiber Optic O_2 Meter (PyroScience). Using this sensor, we have examined release as a function of O_2 concentration. These experiments involved deoxygenating a solution of the fluorescence reporter compound **2** with Ar until a certain O_2 level was reached followed by irradiation of the solution and quantification of ligand release by fluorescence spectroscopy. We have compared a range of physiologically relevant O_2 levels starting with 40 mmHg. Significant release is observed starting at ~ 20 mmHg following a 12 min irradiation. We have added a short discussion of this experiment and included the data in the supporting information.

We have also investigated the effect of initial O_2 concentration on uncaging. A range of physiologically relevant initial O_2 levels (5, 10, 20, and 40 mmHg)⁴¹ were established using Ar deoxygenation. The level of 4-methylumbelliferone release depends on both initial O_2 concentration and irradiation time. Notably, efficient release was observed from the samples with an initial O_2 level of 20 mmHg or less after a 12 min irradiation (Figure S4).

Furthermore, the use of Near-IR light to photocleave a phthalocyanine and produce the PDT drug Pc 4 in water via a radical mechanism has been demonstrated in 2014 by Cheng et al. Unfortunately, this work is not referenced in this manuscript. Instead a lesser relevant work from the same group from 2015 is referenced as ... "has received much less attention.¹³"

That is actually right. It is somewhat of an awkward oversight of this manuscript that this reference is not properly implemented in the introduction of this topic:

"Near infrared light- triggered drug generation and release from gold nanoparticle carriers for photodynamic therapy" Cheng, Y.; Doane, T. L.; Chuang, C. C.; Ziady, A.; Burda, C. Small (2014), 10(9), 1799-1804.

The paper by Cheng et al is not cleaving Si-O bonds but Si-C bonds but the results and the underlying photophysics are strikingly similar.

The issue of the SiPc-carbon cleavage reaction is an important point. Prior studies on cleaving SiPc-C

bonds should have been given more explicit attention in the introduction, although a key reference was included in the original submission.

Of note, while the reaction is similar at the structural level, the underlying photophysics and reaction mechanism are actually quite different. We have clearly shown a dependence on both O₂ and reductant in the SiPc-O cleavage reaction. This is not the case with SiPc-C, which has been proposed to cleave through a radical homolysis mechanism (see reference 17). The dramatic disparities between these two methods are evident from the large difference in estimated bond dissociation energy (~90 kcal/mol for SiPc-O vs ~40 kcal/mol for SiPc-C, reference 17).

We have added three additional key references and the sentences below to fill out this discussion.

The cleavage of high energy SiPc-carbon bonds (BDE ≈ 40 kcal/mol),¹⁷ first observed in synthetic studies, was recently used to cleave the SiPc chromophore from a gold nanoparticle.¹⁸⁻²¹ The focus of these studies is the cleavage of more stable SiPc-oxygen bonds (BDE ≈ 90 kcal/mol),¹⁷ which was only investigated in a single prior study.¹⁴

Thus, the novelty of this work lies mainly in the switching between uncaging and photosensitizing. As said above to make this then worthy of Nature Comm. The quantification and ratios of the two monitored outcomes needs to be presented.

As mentioned above, this is excellent feedback. We have added chemical experiments (detailed above) that provide this quantification. I would note that our biological studies had already illustrated a conversion between two distinct biological mechanisms of action using a combination of cytotoxicity, photopatterning, and FACS-based cell cycle studies. As described below in response to reviewer 3, we have also added experiments that illustrate this conversion over progressively lower O₂ concentrations.

In addition, the work by Lo et al is cited as example for 1,3-bis(trimethylammonium)-compounds, but in the corresponding reference 30, it turns out that the molecules containing these cationic ligands are less effective compared to other ligands. Is that also true for the presented compounds? What is the singlet oxygen yield of each in normoxic conditions. The values are not presented in this manuscript. From the graph on page S6 they seem to be between 5-10%?

On line 106 of the main text, the authors claim that "The quantum yields of ¹O₂ generation (Φ_{Δ}) of 2, 3, and 4 are similar to those of other SiPcs, when measured in an aqueous

environment.²⁵ This is hard to judge in this qualitative comparative statement. Usulan et al. have obtained 11%. The authors should also provide numbers.

Singlet oxygen quantum yields (Φ_{Δ}) were presented in the original submission in Figure 2. As described above in response to reviewer 1, those values have been re-measured using another method and the numbers are still presented in Figure 2. Our numbers are in line with those of other SiPcs.

One challenge for the reader is that 50J seems to be a lot of light energy for a test tube reaction, and one might ask if this hints to a low photoreaction quantum yield - inefficient release of Z-CA4?

The quantum yield is presented in line 129. As mentioned above in response to reviewer 1, our value represents an improvement over recent reports in the green range.

We have also added an experiment to the supporting information in which biological effects resulting from light doses ranging from 0 to 50 J/cm² are examined. Significant effects on cell viability are observed from Z-CA4 release at a light dose of 10 J/cm². We have added a sentence to that describes this experiment and included the data in the supporting information.

We have investigated the role of light dose on these experiments. Notably, significant cell viability effects are observed upon irradiation of 3 under hypoxic conditions with 10 J cm⁻² of light (Figure S10).

From Figure 4 it seems that that Z-CA4 is not photoactive but just a cell toxin. Why then not just deliver Z-CA4, which seems water-soluble, not directly?

The purpose of this approach is to release the bioactive species site-specifically using near-IR light, thereby mitigating systemic toxicity of the chemotherapeutic drug. While targeted drug release is a long-standing goal of cancer treatment, this point should have been made more clearly in the introduction. I have added a short sentence to address this point.

“One appealing alternative is to use near-IR light to deliver or uncage bioactive molecules, thereby generating a spatially targeted biological effect while avoiding undesired side effects associated with systemic delivery.”

There is a typo in "discreet" on line 158.

Modified.

In summary, there needs to be a more quantitative approach to the description of "Uncaging or Photosensitizing". References and the introduction need to clarify that PDT reagents (such as Pc 4) have already been released in water, as cited above. The work by the Knoer group is a good standard of how much singlet oxygen formation one should expect. It should be highlighted.

We appreciate this suggestion, references 33, 39, and 40 include relevant citations with phthalocyanine singlet oxygen quantum yields, including one by Knoer and coworkers.

The paper needs to be accordingly improved in the presentation of the introduction and the results section and then reconsidered for publication.

We are very grateful to this reviewer for these substantive comments, which we believe we have addressed thoroughly.

Reviewer #3 (Remarks to the Author):

The manuscript presents interesting near IR activated drug (CA4)-phthalocyanine system whose mechanism of action, as either a photosensitizer (singlet oxygen) or a prodrug (releasing active drug), depends on oxygen tension. While the photosensitizing mechanism of silicon phthalocyanine under normoxic condition is well known, the drug release from the conjugate by near IR illumination under oxygen-deprived condition is unique, in particular in cultured cells. Although the Si-O bond cleavage chemistry upon light illumination was reported, its application for NIR-activated drug release moves this chemistry one step forward toward future clinical application. Most of experimental designs are sound and well executed. However, following points need to be addressed before recommended for publication.

1. The authors claim this conjugate system is expected to release CA4 (drug) under the hypoxic condition in tumor. They indeed demonstrated it in test tubes and cultured cells. However, it is not clear whether it really will work as they expect in real tumors because the conditions they used (e.g., oxygen-deprivation condition, (extreme hypoxic condition)) may not close to tumor hypoxia. Quantitative definitions and measures need to be used about "oxygen tension" and "hypoxia" [J Natl Cancer Inst (2001) 93 (4): 266-276]. It will be much clearer if they demonstrate it under oxygen partial pressure of hypoxic tumors. Alternatively, in vivo demonstration of drug release in hypoxic tumor will address this question.

This is really an excellent point. While we used the operationally simple, nearly anoxic conditions obtained by streaming N₂ into a plastic chamber, the hypoxia encountered intratumorally is perhaps best captured with 1 or 3% O₂ (such concentrations replicate tumor hypoxia as defined in the paper mentioned above). Using a standard humidified 37 °C incubator set to 1 and 3% O₂ (5% CO₂, balance N₂), we carried out such studies using the photopatterning assay to analyze small molecule release. With both 1 and 3% O₂ significant cellular effects were observed in the region not exposed to light with the caged active compound, but not with caged inactive species. This result suggests that uncaging could occur at the low O₂ levels found in physiological environments. A figure has been added to the supplementary information and the following text to the manuscript reflecting these results.

The photopatterning effect was investigated under varied levels of initial O₂. With both 1% and 3% O₂, cell growth inhibitory effects were observed in the unirradiated area with **3**, but not **4**, albeit somewhat less completely than above (Figure S13). These results suggest that uncaging can occur under the hypoxic conditions found in physiological settings.^{41,51}

Minor points

2. First page: The term, "targeted", may confuse readers. It seems redundancy because in general focused NIR light itself is used.

I agree that this may have been confusing, I have modified the first sentence of the paper to read as follows, which removes the somewhat redundant use of targeted.

"The development of methods that use tissue penetrant near-IR light to treat diseases such as cancer is a long-standing goal."

3. Line 34: It is not clear what does "these approaches sacrifice the benefits of conventional PDT" mean and how their new approach would solve this problem.

I believe the uncertainty here was purely a writing issue. The distinction being made here is between site-specific drug delivery using light as release mechanism and conventional PDT. The point being made here is that it would be advantageous to do both at once; PDT in normal O₂ and small molecule uncaging in hypoxia, which is what this new approach accomplishes. This point is revisited in detail in the conclusion, and I have somewhat rewritten the introduction to clarify the confusing aspect of the original sentence.

"However, only releasing bioactive molecules sacrifice the benefits of conventional PDT treatments (i.e. catalytic light-driven generation of toxic ROS)."

4. Line 205: It is not clear what does "maximal cellular localization" mean. Does it mean maximal uptake to cells or specific subcellular localization to specific organelles in cells?

This simply means the time point at which the greatest amount of cellular signal is observed.

We meant it simply to mean the time point where the greatest amount of intracellular signal is observed. We have rewritten this section to reflect this and to clarify that lysosomal localization is observed.

"Fluorescence microscopy in HeLa cells indicated that these molecules were taken up into cells and appear to localize in lysosomal compartments. Maximal cellular signal was observed following a 3 h preincubation, therefore these conditions were used throughout the studies below (Figure S9)

One final small note. In line 153 of the submitted to manuscript we state the absorbance peak at ~585 nm was distinct from that of a triplet absorbance. While this is true, such triplet absorbance spectra are nearly always derived from transient spectroscopy. Consequently, this statement could be confusing to certain readers. We simply deleted this one statement, which has no impact on the general conclusion that the sharp ~585 nm peak is highly diagnostic of a SiPc radical anion.

REVIEWERS' COMMENTS:

Reviewer #1 (Remarks to the Author):

The changes to the manuscript made in response to the original comments are sufficient. Publication is recommended.

Reviewer #2 (Remarks to the Author):

The authors have done an exemplary job to answer the questions of the reviewers. I have been impressed enough to now suggest this manuscript for publication in Nat. Comm.

Reviewer #3 (Remarks to the Author):

The authors responded reasonably well for all the reviewers' comments. Thus, it is recommended for publication in Nat Com.